# Experimental Study on Damage Evolution Characteristics of Concrete under Impact Load Based on EMI Method

**Bin Tan**

School of Transportation and Logistics, Southwest Jiaotong University, Chengdu 610031, China; tanbin_1989@foxmail.com

**Abstract:** In this paper, the damage evolution characteristics of C50 concrete under impact load were studied, based on the electro-mechanical impedance (EMI) technique. A parallel test was conducted based on the internal relationship between EMI technology and a resonant frequency test to verify the sensitivity and accuracy of EMI technology. In addition, another mechanical test was conducted on specimens with different levels of damage to establish the empirical relationship between the degree of damage and the mechanical properties of concrete. The degrees of damage were discussed by calculating the root mean square deviation (RMSD) index. Results illustrated that the damage changes of concrete can be well monitored by PZT patches. Based on the evolutionary characteristic of the RMSD index, worrying information can be obtained before the destruction of the concrete. On the other hand, mechanical test results indicated that the value of the RMSD was correlated with the splitting tensile strength of the concrete specimens; therefore, it can be used as a reference to evaluate and predict the performance of concrete.

**Keywords:** EMI technology; resonance frequency test; damage evolution; splitting tensile strength; risk alert

## 1. Introduction

As a quasi-brittle material, the performance of concrete is prone to degrade under repeated impact loads. For example, both the primary support and the secondary lining of mountain tunnels inevitably suffer various degrees of blasting impact loads during the construction process. With the evolution and connection of micro-cracks, concrete damage accumulates gradually under the impact load until visible cracks suddenly appear. After this, the bearing capacity of the concrete structure will drop sharply, and structural safety is hard to guarantee; therefore, it is of great engineering significance to adopt effective methods for monitoring the damage evolution of concrete under impact loads [1].

As a new health monitoring technology, the EMI method has come into public view since the mid-1990s. Owing to a high working frequency and high sensitivity to slight damage, the EMI method is applicable to many engineering fields. Consequently, by raising widespread concern from academic and engineering fields, a large number of experiments and tests were carried out.

Sun et al. [1] conducted an electrical admittance test of combined truss structures with different degrees of damage. The EMI method was first used to monitor the health of engineering structures. The relative deviation (RD) was defined as the index of damage degree and was used for detecting the damaged location. Test results showed that the EMI method performed well in evaluating the condition of truss structures. Ayres et al. [2,3] carried out further investigations on the application range of the EMI method. A large laboratory experiment was conducted on a steel bridge using the EMI method, in which the degree of damage was measured by the number of loose bridge bolts. The results proved that the EMI method is applicable not only on light structures, but also on heavy structures, such as a steel bridge. Giurgiutiu et al. [4] tested fatigue damage on structural joint welds, using the EMI method. The damage index was obtained by Euclid norm,

according to the impedance spectrum signal. Results showed that the damage index has a strong correlation to the number of weld cracks, which verified the sensitivity of impedance signals towards weld fatigue damage. Based on the former experimental results, Giurgiutiu established the relationship between the damage index and the remaining life of structures. Researchers from Nanyang Technological University (Singapore) extended the EMI method to reinforced concrete structures. During a failing test, Soh [5] utilized the EMI method to monitor the damage evolution characteristics of a reinforced concrete bridge. The high-frequency electrical admittance signals of the bridge under different loads were collected, and the root mean square deviation (RMSD) was used as an indicator of the degree of damage. Results showed that the EMI method still has a high sensitivity for damage detection of reinforced concrete structures, and has the ability to perceive the emergence and development of structural defects. Bhalla et al. [6] placed several PZT (Piezoelectric Zirconate Titanate) patches at the beam-column joint, and middle of the beam bottom, of a two-story reinforced concrete frame structure. The vibration platform test and electrical impedance signal test were performed, simultaneously. They proposed a new concept named "Effective Impedance", an original detection method for structural damage. Tseng et al. [7,8] employed the impedance method to detect the deterioration of concrete caused by a corrosive environment of sulfate and chloride. The result indicated that the EMI method is able to identify the degradation of concrete caused by chemical corrosion. The author's team [9–12] employed the EMI method to study fatigue damage of the tunnel structure. The RMSD value of the electrical admittance signal was used as an indicator of the degree of damage. The correlation between the RMSD and cyclic load was established on the basis of the test results. In addition, a fatigue damage model for the bottom of the tunnel structure was proposed. The research proved that the EMI method is feasible to detect the damage of underground structures.

As can be seen from the aforementioned literature, the EMI method has a wide feasibility over various kinds of structure, material, and damage [13–15]. However, research around the use of the EMI method in civil engineering is still limited, compared to aeronautical and mechanical engineering, especially in regard to the aspect of monitoring concrete structural performance [12,16,17]. Existing experimental research mostly uses prefabricated cracks to simulate the damaged state, and analysis mainly focuses on the comparison of the electrical admittance signals between the final damaged state and the initial state. However, damage evolution characteristics of the concrete materials are equally important, especially for those under impact load [18,19]. Note that C50 concrete (standard compressive strength of 50 MPa) has been widely used in protective engineering and civil engineering. For example, C50 concrete is usually used to line tunnel segments. It is also used in pile foundations of high-speed railway bridges [20,21]. Thus, it is critical to monitor and understand damage evolution characteristics of C50 concrete materials [22,23].

The main purpose of this presented work is to propose a new procedure to monitor the damage state, and predict the mechanical properties of concrete without disturbance and destruction, which helps to increase the sustainability of existing concrete structures. The author used the EMI method to record impedance signal changes in C50 concrete specimens from integrity to destruction, under the impact load. Using a resonant frequency tester, a parallel test was conducted to verify the results of the EMI experiment. Thus, the effectiveness and sensitivity of the EMI method on monitoring the impact damage evolution of concrete was discussed. A mechanical strength test was conducted to investigate the relationship between the mechanical properties and the degree of impact damage of the concrete specimens, which can further provide reference for relevant research.

## 2. The Principle of Parallel Tests

According to the classical electromechanical one-dimensional coupling model (Figure 1) introduced by Liang et al. and Su et al. [24,25], a spring-mass-damping system (which considers one degree of freedom) was used to describe the dynamic response of PZT sheets attached to the surface of the structure in an electronic field.

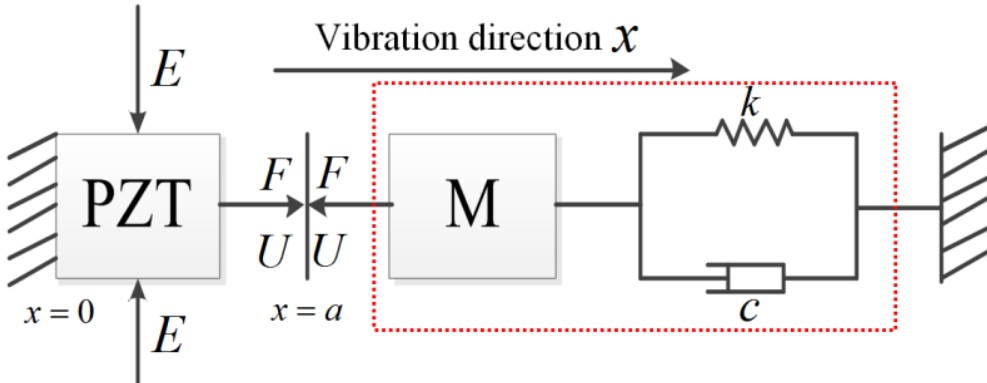

**Figure 1.** One-dimensional vibration theory model for PZT.

Under the action of an alternating electric field, considering the vibration of the PZT film along the direction $x$, the one-dimensional vibration equation of the PZT film is obtained by:

$$\rho \frac{\partial^2 u(x,t)}{\partial t^2} = \overline{Y}_{11}^E \frac{\partial^2 u(x,t)}{\partial x^2} \tag{1}$$

where, $u(x,t)$ is the displacement of PZT in the direction of $x$; $\rho$ represents the density of PZT; $\overline{Y}_{11}^E$ is the complex Young's modulus of PZT.

The boundary condition of PZT patches is mechanically free, and it is electrically shorted. The relevant constitutive equation is:

$$\begin{aligned} \varepsilon_1 &= \overline{c}_{11}^E \sigma_1 + d_{31} E \\ D_3 &= \overline{\varepsilon}_{33}^\sigma E + d_{31} \sigma_1 \end{aligned} \tag{2}$$

where, $\varepsilon_1$ represents strain; $D_3$ represents electrical displacement; $\overline{c}_{11}^E$ is a short-circuit complex elastic compliance coefficient; $d_{31}$ is a piezoelectric strain constant; $\overline{\varepsilon}_{33}^\sigma$ is a complex dielectric constant; $\sigma_1$ represents stress.

The Equation (1) can be solved by using the mechanical balance and displacement coordination relationship while $x = 0$ and $x = a$. Thus, the equation for the one-dimensional displacement of PZT can be obtained.

Combined with the constitutive Equation (2) of PZT and the definition of electric admittance, the simplified expression for coupled electric admittance of PZT, which is in the low frequency range, can be established by:

$$Y = j\omega \frac{ba}{h} \left( \overline{\varepsilon}_{33}^\sigma - \frac{Z_s}{Z_a + Z_s} d_{31}^2 \overline{Y}_{11}^E \right) \tag{3}$$

where, $Y$ is the electric admittance of PZT; $a$, $b$, $h$ are the length, width and thickness of PZT patches, respectively; $\omega$ is the excitation frequency; $j$ is the imaginary unit; $Z_s$ and $Z_a$ are the mechanical impedance of structure and PZT, respectively. The specific expressions are:

$$\begin{aligned} Z_s &= c + m \frac{\omega^2 - \omega_n^2}{\omega} i \\ Z_a &= -\frac{K_a(1+\eta j)}{\omega} \frac{ka}{\tan(ka)} j \end{aligned} \tag{4}$$

where, $c$ is the structural damping; $\omega_n$ is the natural frequency of the structure; $K_a$ is the static stiffness of PZT; $k = \omega/c_1$ represents the wave number in the range of $2\pi$; $c_1 = \sqrt{Y_{11}^E/\rho}$ is the wave velocity of PZT.

According to the linear vibration theory, in a single-degree-of-freedom vibration system, the relationship between acceleration resonance frequency, velocity resonance frequency, and the natural frequency of the system, are obtained by:

$$\omega_a = \omega_n \frac{1}{\sqrt{1 - \frac{c^2}{(4m\pi)^2 \omega_n^2}}}; \quad \omega_v = \omega_n \tag{5}$$

Thus, the damping expression can be expressed as:

$$c = 4m\pi \frac{\omega_v}{\omega_a} \sqrt{\omega_a^2 - \omega_v^2} \tag{6}$$

By using Equation (6), the expression of mechanical impedance in Equation (4) can be rewritten as:

$$Z_s = 4m\pi \frac{\omega_v}{\omega_a} \sqrt{\omega_a^2 - \omega_v^2} + m \frac{\omega^2 - \omega_v^2}{\omega} i \tag{7}$$

It can be seen from the combination of Equations (3), (4), and (7) that the change in the PZT's electric admittance is only related to the mechanical impedance of the structure itself, under the premise that the PZT's own performance remains stable. Thus, the change in structural impedance can be attributed to the change in the resonant frequency of the structure, which can be conveniently monitored by the mechanical impedance of the PZT patches. However, once the PZT patches are fractured during the impacting process, damage to the monitoring objects cannot be measured appropriately.

The internal damage of the concrete gradually accumulates under the long-term impact load, while the stiffness of the concrete decreases simultaneously. The resonance frequency of the concrete is then bound to shift to a different frequency. The deviation degree of resonance frequency determines the variation of the structure's mechanical impedance directly, which can be reflected by the electrical admittance signal. Thus, the sensitivity and accuracy of the EMI test results can be reflected from another aspect, through the resonance frequency test. In real applications, the PZT patches can be conveniently attached to the surface of the monitoring object to evaluate the damage state.

In the present study, a parallel test was designed to test the resonance frequency of the concrete specimen, based on the internal relationship between the EMI method and the resonance frequency test, so that the test results of the EMI method could be verified and supplemented. Furthermore, based on regression analysis, the tensile splitting strength test of the concrete structure, and the quantitative correlations between the impact damage and the tensile strength of concrete, were obtained.

### 3. Design of the Test

#### 3.1. The Specimen and the Test Equipment

Table 1 shows the mixture components of C50 concrete, which is commonly used in tunnel lining structures in China. Each specimen was a prism, with a length and width of 100 mm, and a height of 300 mm. Additionally, the strength of the specimens was guaranteed by preparing the three standard specimens for the uniaxial compressive strength test. During specimen preparation, sand, cement, and rock blocks were poured in to a mixing machine sequentially, and were stirred for 30 s. Then, water was moderately added and stirred well. The inner wall of the mold was consistently coated with mineral oil. The mixture was poured in to the mold and subjected to mechanical vibration for 20 s. A steam machine, with an accurate and automatic controlling system, was adopted to guarantee a consistent curing process. The steam curing process is shown in Table 2. The specimens were demolded after 24 h, then cured in a specialized laboratory with a stable temperature of 25 °C, and relative humidity of 95%, for 90 days. The generated specimens were, therefore, almost consistent in mechanical properties, with very limited variation and uncertainty. The initial mechanical parameters of the specimens are shown in Table 3.

**Table 1.** Mix proportion of concrete.

| Components | Mass (kg/m$^3$) |
|---|---|
| Cement (42.5 MPa) | 380 |
| Fly ash (classI) | 100 |
| Fine aggregate (Medium sand) | 692 |
| Coarse aggregate (5–25 mm continuous grading) | 1128 |
| Water | 150 |
| Water reducing agent (solid polycarboxylate acid) | 0.5 |
| Specimens' size | 100*100*300 mm |
| Compression specimens' size | 100*100*100 mm |

**Table 2.** Steam curing system.

| Rest Curing Time (h) | Heating Rate (°C/h) | Constant Temperature Time (h) | Constant Temperature (°C) | Cooling Rate (°C/h) |
|---|---|---|---|---|
| 3 | 15 | 3 | 50 | 15 |

**Table 3.** Mechanical parameters of the specimens.

| Dynamic Elastic Modulus (MPa) | Shear Modulus (MPa) | Compressive Strength (MPa) | Poisson's Ratio |
|---|---|---|---|
| 56.3 | 23.6 | 52.4 | 0.23 |

### 3.2. Experimental Scheme

Figure 2 shows the test procedure of the EMI method. In the test, shown in Table 4, impact loads with different energies were obtained by changing the drop height. During the tests, the specimens are placed horizontally and freely on a rubber plate with a thickness of 10 mm.

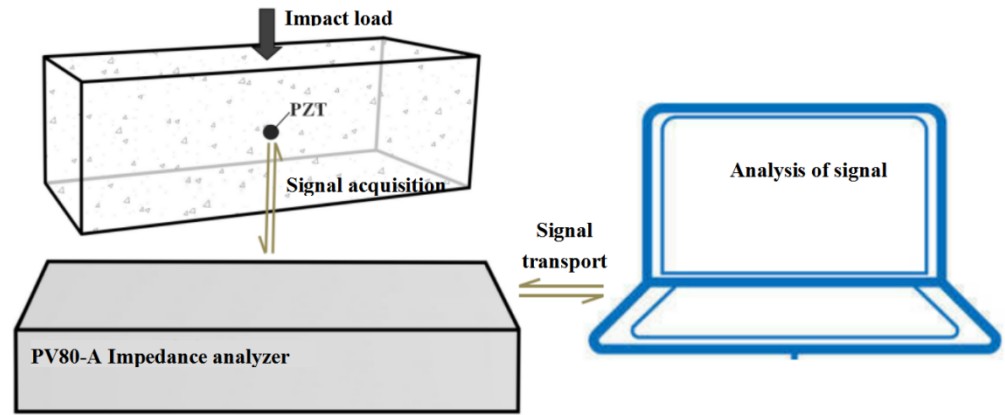

**Figure 2.** Flow chart of the EMI test.

**Table 4.** Test conditions.

| Group | Number of Specimens | Impact Height/cm | The Mass of the Punching Ball |
|---|---|---|---|
| A | 4 | 20 | |
| B | 4 | 25 | 4 kg |
| C | 4 | 30 | |

### 3.3. The Test Equipment

The PZT-5 patches were used to monitor the damage state of the concrete specimens, which were extremely sensitive to the drive signal. Each PZT-5 patch was a sheet with a di-

ameter of 15 mm and a thickness of 0.5 mm. Through additional work by the manufacturer, the two electrodes of the PZT-5 patches were on the same side, so that it could be pasted closely to the concrete surface, as shown in Figure 3.

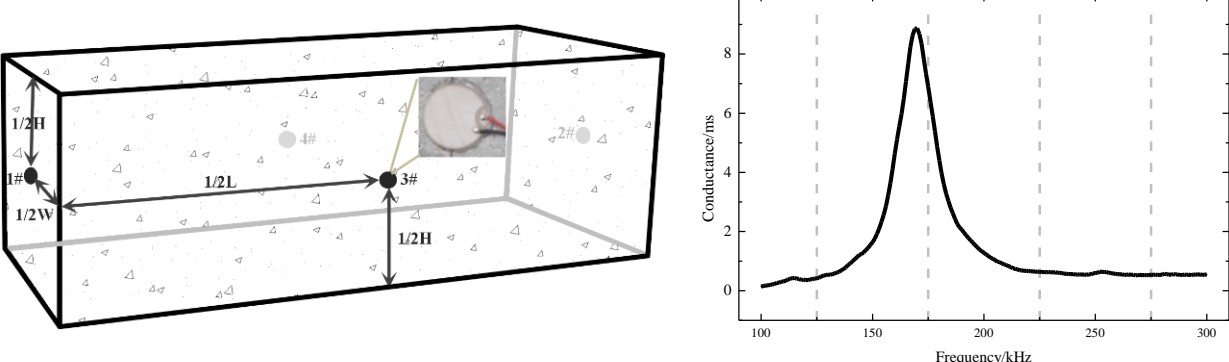

(**a**) Positions of PZT patches on the specimen　　　(**b**) The conductance curve for the free PZTs

**Figure 3.** Basic information about the PZT patches.

The patches of the PZT-5 were attached symmetrically, on both ends (Patches 1# and 2# are shown in Figure 3a) and in the middle of the specimen (Patches 3# and 4# are shown in Figure 3a), as shown in Figure 3. Therefore, four sets of test data could be obtained from the two prepared specimens for each experimental condition. Thus, information about the damage state of the concrete specimens could be perceived comprehensively through the electrical admittance signal from different locations. This provides basic data for further analysis. The adhesive between the PZT patches and the specimens was AB glue.

A PV80-A impedance analyzer, with sweep frequency ranging from 1 kHZ to 1 MHZ, was used to collect the piezoelectric signal. It was validated that the PZT patches become sensitive to their own conditions, rather than the conditions of the structure they are bonded to, when excitation frequency is higher than 500 kHz, which is much higher than the first resonant frequency [26–28]. Also, frequencies 100 kHz and 300 kHz were proven to be reasonable ranges to study the damage state of the concrete, according to research work conducted by Liu and Prteek Negi [11,18]. Thus, the excitation frequency during the experiment ranged from 100 kHz to 300 kHz, and the number of sampling points was 200. The conductance curve for the free PZTs is shown in Figure 3b.

An E-Meter MK II resonance frequency tester, with an accelerometer sensitivity of 9.60 mV/g, was adopted to capture the vibration response characteristics of the concrete specimen (Figure 4). The vibrating accelerations in the time domain can be recorded directly using the E-Meter MK II resonance frequency tester. Then, the vibration accelerations in the time domain can be transformed into the frequency domain by Fourier analysis. Therefore, acceleration resonance frequency can be easily obtained, which is the frequency with maximum amplitude. On the basis of the operator's manual [24], the acceleration resonance frequency was automatically converted into dynamic elastic modulus by:

$$E_{dy} = \lambda \times \frac{H \times M}{L \times W} \omega_a^2 \tag{8}$$

where $E_{dy}$ is the dynamic elastic modulus; $\omega_a$ is the acceleration resonance frequency; $H$ is the height of the specimen; $L$, $W$ is the length and width of the specimen, respectively; $M$ is the mass of the specimen; and $\lambda$ is the system constant, which is calibrated to 5.093 for a cylinder specimen, and 4.000 for a prism specimen [24].

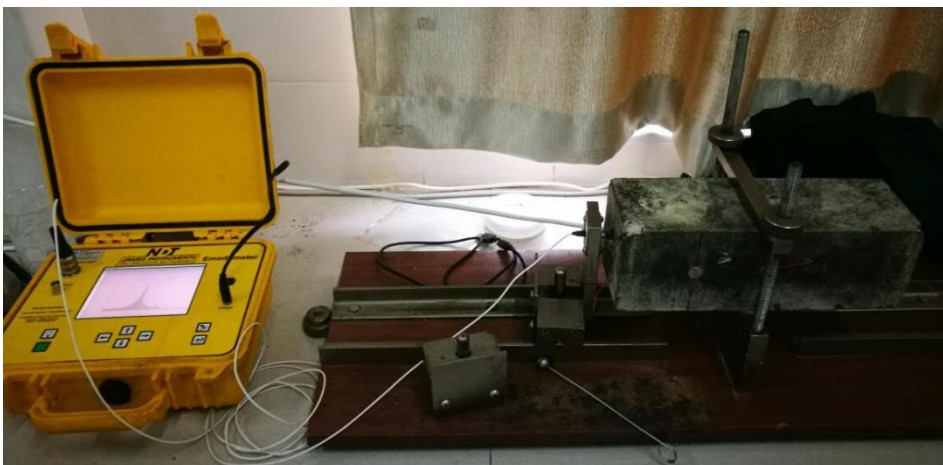

**Figure 4.** E-Meter MK II resonance frequency tester.

## 4. Analysis of Test Results

### 4.1. Analysis of Piezoelectric Signals

Figure 5 shows experimental results of the EMI method and the resonant frequency test of group-A. On the basis of the evolution characteristics of the piezoelectric signal and dynamic elastic modulus, it shows that the damage caused by the impact on the concrete specimens increased gradually during the early stage of the impact experiment. It is inferred that the inner micro-cracks accumulated and developed in this period. In the end, visible cracks suddenly appeared when the damage accumulated to a certain degree. Simultaneously, the piezoelectric signal changed abruptly, along with a sharp deduction of the peak values of signal, as shown in Figure 5a. In addition, the mechanical properties of the specimens decayed dramatically in this period, which is consistent to the conclusion of the previous study [11].

As shown in Figure 5, the attenuation law of the dynamic elasticity modulus was highly consistent with the change in piezoelectric signals, which confirmed that the EMI method was effective in capturing the development process of the specimen's damage.

However, it was noticed that the peak values of the conductance signal did not drop sharply, as shown in Figure 5c. Comparing the crack positions of specimens I and II, there was a relatively long distance between them and the PZT-5 patches, as shown in Figure 5c. In addition, the cracks of specimen II did not penetrate through the specimen, as they did in specimen I. These are the main reasons for the limited changes in conductance signal of specimen II. In addition, the anisotropy and heterogeneity of the concrete contributed to the diversities of the conductance test results, to some extent.

In order to quantitatively study the damage development characteristics of the specimens before failure, appropriate indicators are supposed to be employed to describe the damage of the concrete specimens.

Previous literature [9–11] suggests that the root mean square (RMSD) can well reflect the change in structural damage. The corresponding expression is as follows:

$$\text{RMSD} = \sqrt{\frac{\sum_{i=1}^{N} \left(G_i^n - G_i^0\right)^2}{\sum_{i=1}^{N} \left(G_i^0\right)^2}} \tag{9}$$

where, $G_i^0$ represents conductance value of the concrete specimens before impact, which is often defined as a non-damage reference; $G_i^n$ is the conductance value of the structure after $n$ times of impacts. $N$ is the total number of signal sampling points.

In the present study, the RMSD was used as an indicator to offer quantitative analysis of the damage evolution characteristics of concrete specimens.

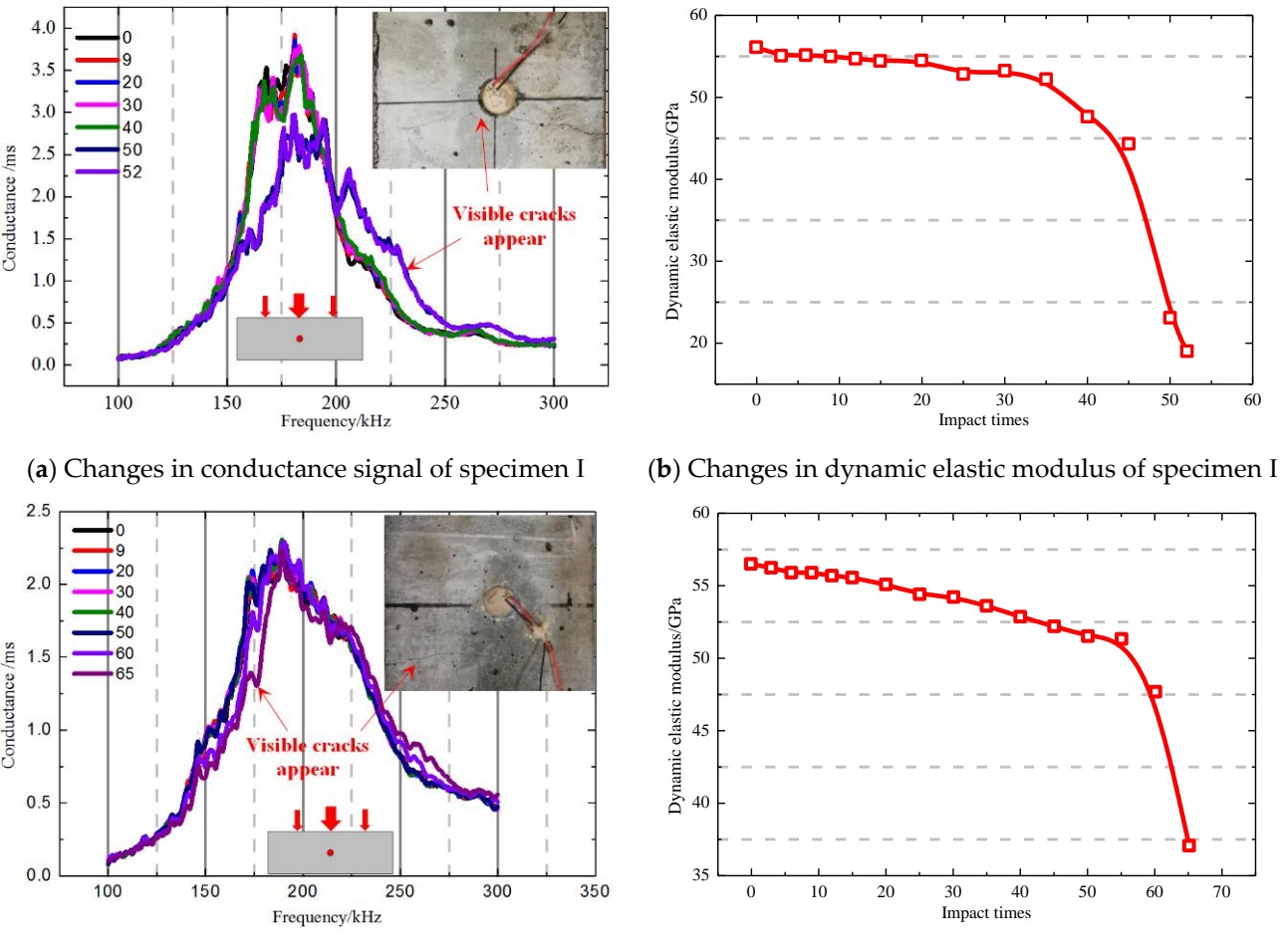

(**a**) Changes in conductance signal of specimen I

(**b**) Changes in dynamic elastic modulus of specimen I

(**c**) Changes in conductance signal of specimen II

(**d**) Changes in dynamic elastic modulus of specimen II

**Figure 5.** Changes in conductance signal under different impact times.

According to the experimental results of Group A (Figure 5), the development process of damage caused by impact can be generally divided into three stages. In the early stage of damage evolution, internal micro-cracks emerge rapidly as the impact load continues. The internal structure of the specimens, such as porosity, was reconstructed in this period, which lead to the obvious change in piezoelectric signals. With the development of internal cracks, flexibility of the specimen gradually increased, as did buffering capacity. Thus, damage of the specimen grew slowly and stably. However, the damage still accumulated gradually, and micro-cracks continued to develop and connect to each other. Finally, visible cracks appeared suddenly, and the value of the piezoelectric signals dropped sharply; therefore, it can be stated that the evolution of the damage caused by impact showed significant nonlinearity.

Figure 6 illustrates that the amplitude of the 3# and 4# piezoelectric signals were obviously higher than those of the 1# and 2# piezoelectric signals. The degree of damage in the middle of the specimen was higher than that at the end of specimen because of the way the damage had been fabricated. Results verified that the change in piezoelectric signals were not only related to the degree of damage, but also to the distance between the pasting position and the damaged area; the shorter the distance, the more sensitive the signal was, which indicated the validity of the experimental results.

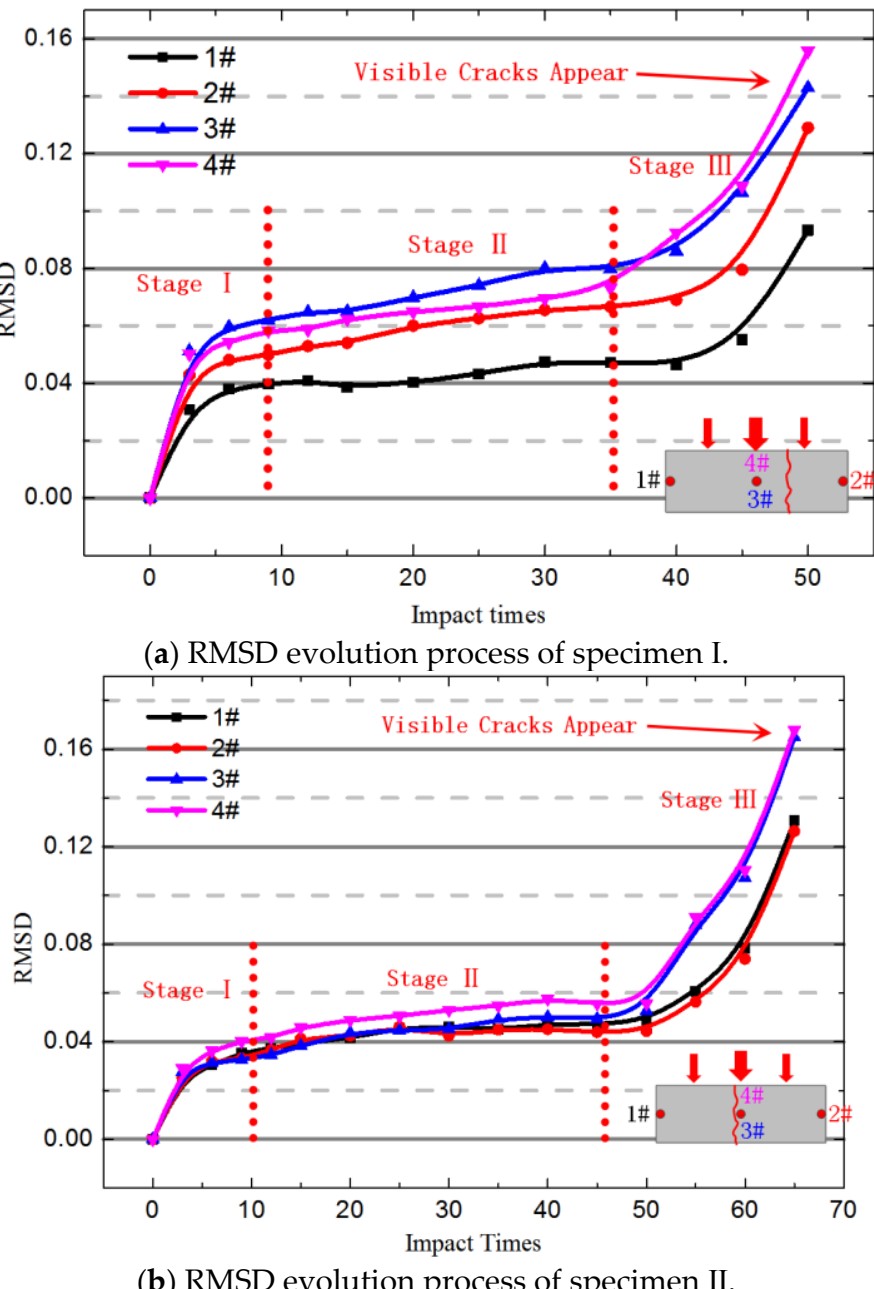

(**a**) RMSD evolution process of specimen I.

(**b**) RMSD evolution process of specimen II.

**Figure 6.** RMSD damage value evolution curve of Group A.

Figure 7 shows the increment values of the RMSD after every five impacts on specimens in group A.

In the early stage of damage emergence, damage of the specimens increased greatly due to the transition from non-damaged state to damaged state. In the middle of the damage accumulation stage, the RMSD increment of the specimens were relatively small. In the final stage of failure, visible cracks were about to appear and the increment of the piezoelectric signals significantly improved, compared with the second stage, which can be used as an alarm signal before specimen failure. When visible cracks appeared, the increment value of the damage reached its maximum value. The concrete specimen had basically lost its function, and it was on the verge of brittle fracture. Thus, this can be used as a sign of failure.

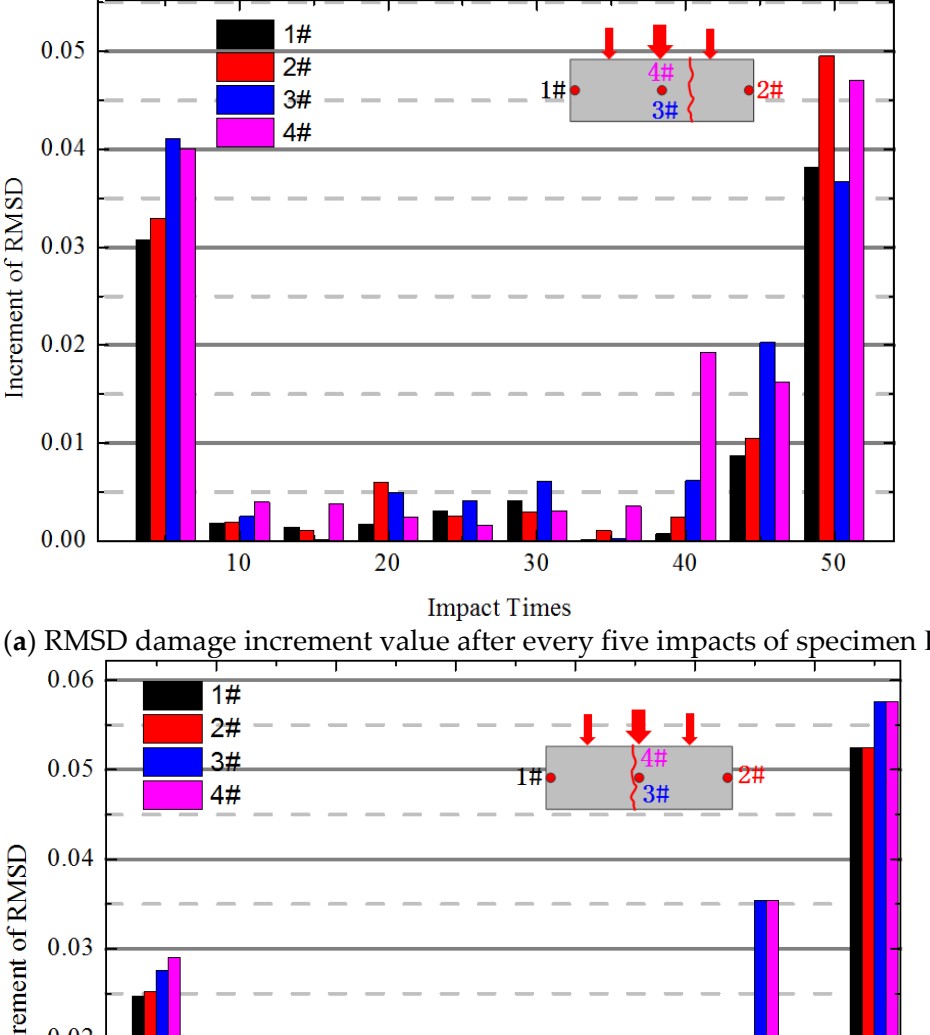

(**a**) RMSD damage increment value after every five impacts of specimen I.

(**b**) RMSD damage increment value after every five impacts of specimen II.

**Figure 7.** RMSD damage increment value distribution of Group A.

Figure 8 shows the evolution process of the damage value of the RMSD for different impact energies, against impact times. As can be seen in Figure 8, damage increment at the early stage increased with the rise of impact energy. The second stage of damage accumulation was observably shortened.

In particular, as shown by Group C in Figure 8, the damage value of the RMSD increased rapidly, until the concrete specimen broke. The stage of damage accumulation barely existed during the condition of high impact energy, and the number of impact times that specimens could bear dropped dramatically, as impact energy increased (Figure 9). In conclusion, as a quasi-brittle material, concrete is extremely sensitive to impact energy, and its structure may be vulnerable when the impact load exceeds a critical threshold.

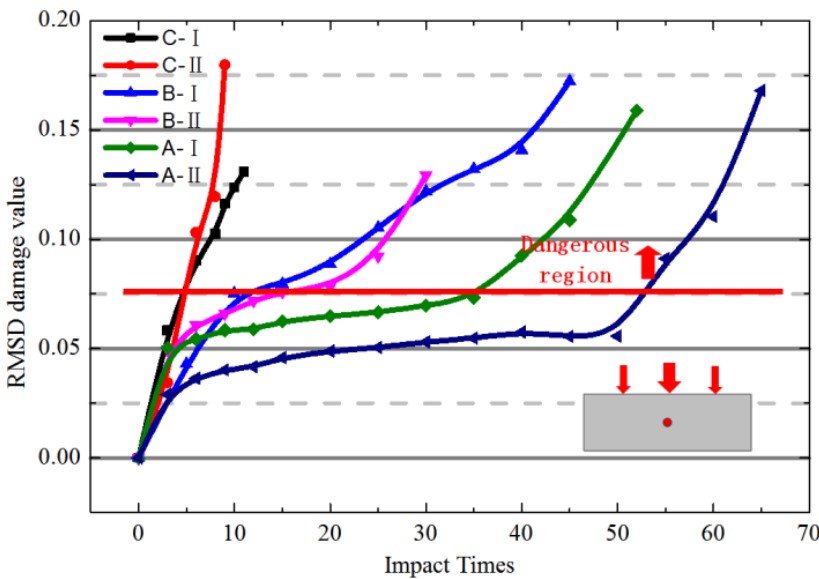

**Figure 8.** Development trend of concrete RMSD damage values, under different impact loads.

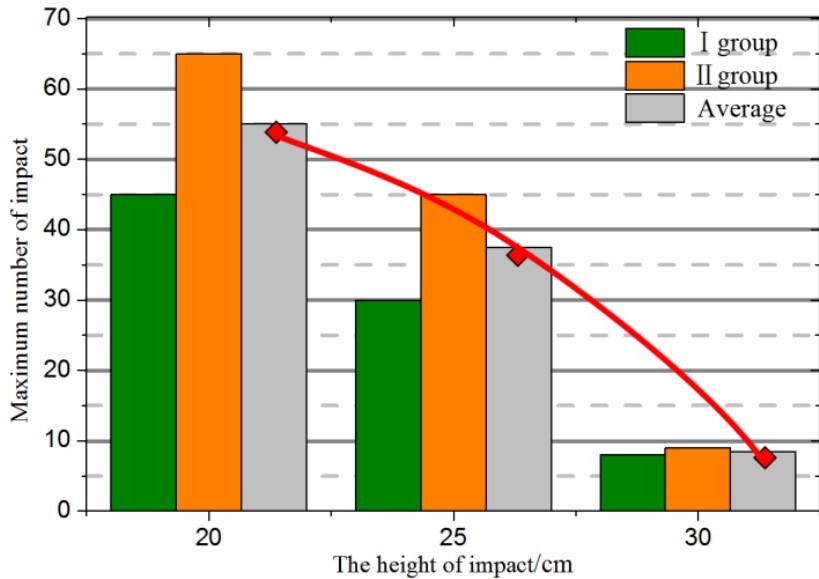

**Figure 9.** Schematic diagram of ultimate impact times of concrete, under different impact heights.

On the basis of the damage evolution characteristics of group A, B, and C, the degree of damage of the concrete specimens developed rapidly when the RMSD value of the specimen exceeded 0.075. The mechanical performance of the concrete specimens was highly likely to deteriorate in the interim. Hence, the RMSD index can be used to define the range of danger in the concrete structure. Experimental results indicate that the RMSD index can be used to effectively define the damage status of plane concrete.

### 4.2. Analysis of Parallel Verification Test Results

As described in Section 1, in order to verify the reliability of monitoring the damage caused by the impact of concrete using the EMI method, a resonant frequency tester was utilized for parallel testing. In order to use the traditional definition of damage (as shown in Equation (10)), the dynamic elasticity modulus was adopted to define the degree of damage to the concrete specimens, with different damage states. The expression is as follows:

$$D = 1 - \frac{E_d}{E_s} \tag{10}$$

where, $E_d$ represents the dynamic elasticity modulus of damaged specimens; $E_s$ represents the initial dynamic elasticity modulus; $D$ is the damage scalar.

Comparing the data in Figures 8 and 10, the evolution characteristics and the turning point of the damage value defined by the RMSD value were consistent with that defined by the dynamic modulus. Hence, it can be stated that the EMI test results, and the dynamic elasticity modulus test results, were different expressions of the same damage state of the concrete specimens. It also proved that the change in piezoelectric signals derive from the change of structural resonant frequency, which can be accurately reflected in the damage state of concrete material. Although noting that there is a thin layer between the patches and concrete formed by the binder, previous researchers have validated that the influence of this small layer is negligible.

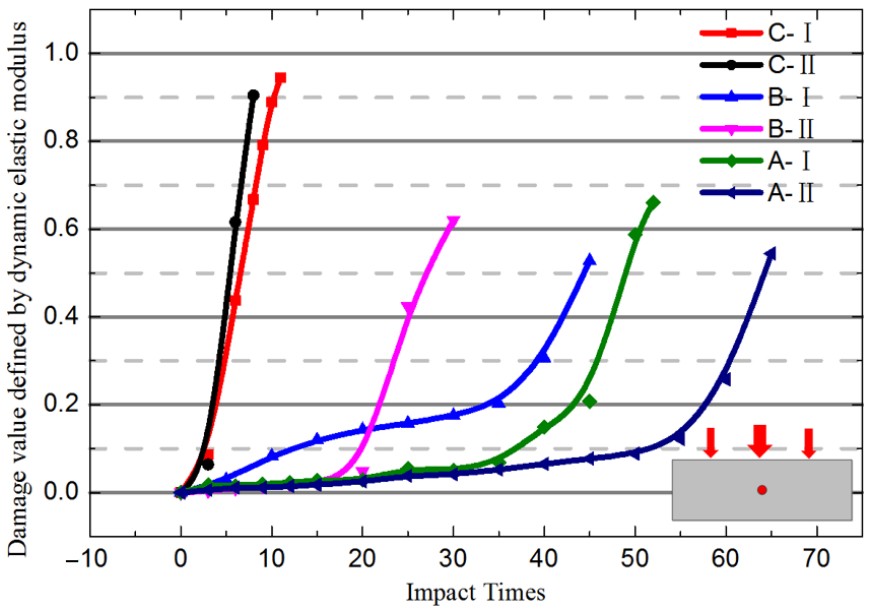

**Figure 10.** Development trend of damage value under different impact loads.

### 4.3. Analysis of the Results of Splitting Tensile Strength

The EMI test results show the damage status of concrete. The mechanical properties of the concrete will be significantly weakened by increasing damage. The splitting tensile strength is a critical index that represents the mechanical properties of the concrete. By establishing the relationships between the EMI test results and mechanical test results, the mechanical properties of the concrete under various damaged states can be predicted. In order to establish the relationship between the non-destructive test signal and mechanical properties of the concrete specimens, nine concrete specimens with different degrees of damage were fabricated, and the split tensile strength test was thus conducted. As shown in Figure 11, three typical sections of each specimen were selected for the split tensile strength test.

According to test results, affected by the degree of damage fabrication, the split tensile strength of one certain section of almost all the specimens was close to that of the concrete specimen with no damage. Because of the inability to show the difference in damage, these data were eliminated from this paper. Two relatively smaller values, of split tensile strength and their mean value, were chosen as the characterization of the tensile strength. The experimental data is shown in Figure 12.

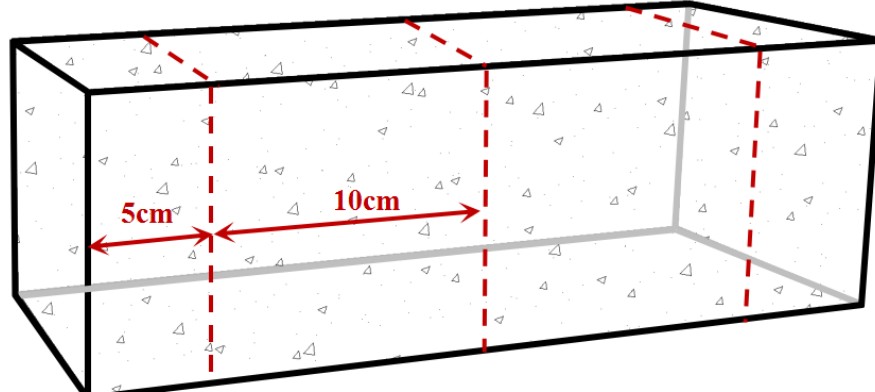

**Figure 11.** Schematic diagram of typical section location.

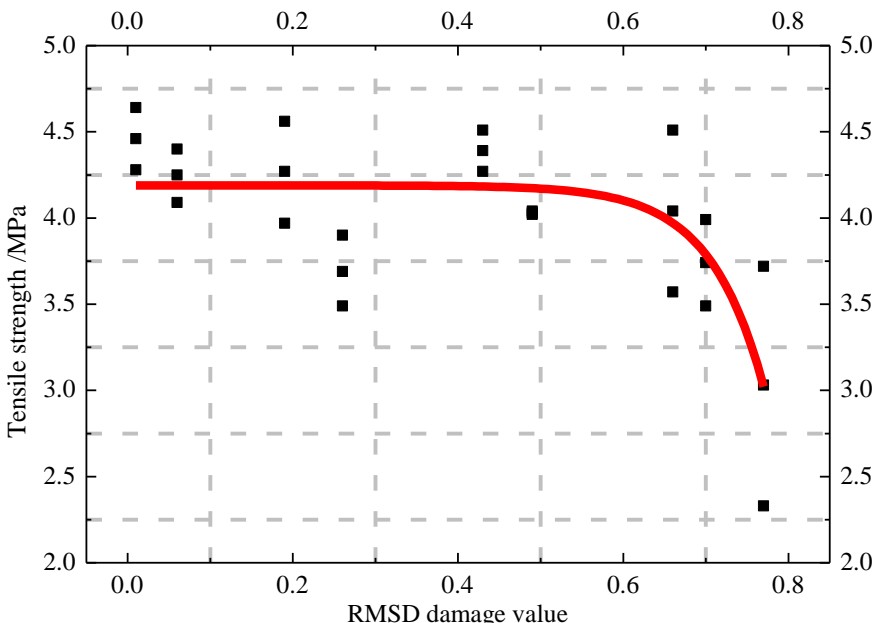

**Figure 12.** Correlation between split tensile strength and the RMSD damage value.

The fitting function in Figure 12 is:

$$f = -2.924E^{-5} \times \exp(-\frac{\text{RMSD}}{0.00688}) + 4.237 \tag{11}$$

where, $f$ is the split tensile strength.

The fitting curve is marked red in Figure 12. It is indicated that the split tensile strength of the concrete specimen exponentially attenuated as the RMSD value increased. Similar to the damage evolution characteristics observed in previous research, the split tensile strength of concrete reduced slowly in the stage of damage initiation and accumulation. In the interim, there was no significant change in mechanical properties of the concrete specimen. When the damage gradually accumulated and exceeded a certain threshold (RMSD $\geq$ 0.075), the concrete split tensile strength may enter the rapid decay stage. At this stage, even slight development of the RMSD damage value may also cause the concrete split tensile strength to reduce significantly, triggering brittle fracture of the specimens. The RMSD damage value has a certain correlation with the split tensile strength of the concrete specimen. Once quantitative relations between the damage state and mechanical strength are established, the proposed procedure be used to predict the mechanical properties of the concrete specimen, without disturbance and destruction.

## 5. Summary

Based on theoretical analysis, the intrinsic relationship between the EMI method and resonant frequency tester was established. Impact damage evolution characteristics of C50 concrete were explored using the EMI method. Through the parallel test of resonant frequency tester, the sensitivity and accuracy of monitoring impact damage evolution using the EMI method were confirmed. Furthermore, the relationship between the RMSD damage value, and mechanical properties, was obtained on the basis of mechanical experiment results.

As results illustrated, the damage evolution process of the specimens can be generally divided into three stages: damage emergence stage, damage accumulation stage, and failure stage. Before the final failure of the concrete specimens, the RMSD value was bound to rise observably. In addition, the standard concrete specimens in this study were highly probable to enter the rapid damage development stage when the RMSD value exceeded 0.075, and brittle fracture occurred easily in the interim. The phenomena mentioned above indicated that the RMSD damage value obtained by the EMI method has observable evolutionary characteristics before the failure of a concrete structure, which can be used as an early warning sign.

According to a parallel experiment, the damage evolution characteristics of the resonant frequency test was consistent with that of the EMI method, which confirmed the reliance of EMI test results. The split tensile strength test indicated that the RMSD damage value, and the split tensile strength of the concrete specimen, satisfied a certain empirical relationship. In addition, the quantitative evaluation and prediction of the bearing capacity of the specimens can be obtained through their empirical relationship.

**Funding:** Project of National Railway Administration (KF2021-36).

**Institutional Review Board Statement:** Not applicable.

**Informed Consent Statement:** Not applicable.

**Data Availability Statement:** The data will be obtained by contacting the corresponding author.

**Conflicts of Interest:** The authors declare no conflict of interest.

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
