# Peer review of "Experimental Study on Damage Evolution Characteristics of Concrete under Impact Load Based on EMI Method"

_sustainability, doi:10.3390/su141710557_

Round 1

Reviewer 1 Report

In this paper, 4 specimens were prepared for each group. However, results and discussion were based on only 2 specimens for each group. Why authors show only results of 2 specimens? Please show the reason in the text.

In the experiment, please show the age of concrete specimens and support conditions of the specimens.

In the equation 8, how the authors obtain or determine the acceleration resonance frequency? Please add detail of the measurement.

In 3.1 section, it is difficult to understand what the authors mentioned. For example, “the impact damage of the concrete specimen increases obviously during the early stage of the impact experiment.” What is the impact damage?

“the inner micro-crack developed gradually and connected to each other.” How did the authors confirm these phenomena?

The end of 3.1 section, “the RMSD index can be used to define the dangerous range of the concrete structure” is not correct. Because concrete structures are usually involved reinforcement, results of this manuscript were based on only plane concrete.

The end of 3.2 section, what is “the bonding layer”?

In 3.3 section, why splitting tensile strength were compared to the results of the EMI test? The reasons must be shown in the text.

Author Response

The detial responses can be found in the attachments.

Reviewer 2 Report

Hello.

Wish you best of luck!

Author Response

The detial responses can be found in the attachment. 

Reviewer 3 Report

Please, see the comments in the attached PDF file 

Author Response

(The authors gave the same response as above.)

Reviewer 4 Report

This article is very interesting and the experimental methodology is well described.

Some modifications are needed before the pubblication.

Introduction paragraph could be improved, enriching the state of the art about the present research issue.

Some details in Figures could be added.

Comments:

-Affiliation: The email is repetead twice.

ABSTRACT:

C50, PZT (Please define each abbreviaiton in the text).

INTRODUCTION: 

Please try to add other references enlarging the state of the art.

Please try to highlight the aims of this research paper.

Please try to underline the link between this research topic and the sustainability macrotopic.

Please add some references about other experimental current previous papers.

LINE 9: "impact loads".

Please here add this reference:

-Kucíková, L, Šejnoha, M, Janda, T, Sýkora, J, Padevetand,P, Marseglia, G. Mechanical properties of spruce wood extracted from GLT beams loaded by fire,Sustainability 2021, 13(10), 5494; https://doi.org/10.3390/su13105494

-Figures 5 and 6, 7,8,10 . Please put the measure units in x-axis in Figures 5b and 5d and 6.

-CONCLUSIONS:

Please underline the point of force of the paper.

REFERENCES:

Please check the referene style.

See for example REFS 12,18,19,20,21 (without abbreviation of the authors name)..

Please add the doi for each Ref.

Author Response

(The authors gave the same response as above.)

Round 2

Reviewer 1 Report

no comment.